# Triazine: An Important Building Block of Organic Materials for Solar Cell Application

**DOI:** 10.3390/molecules28010257

**Published:** 2022-12-28

**Authors:** Valeria Dávila Cerón, Luis Alberto Illicachi, Braulio Insuasty

**Affiliations:** 1Heterocyclic Compounds Research Group, Department of Chemistry, Universidad del Valle, A.A., Cali 25360, Colombia; 2Research Group of Chemical and Biotechnology, Faculty of Basic Sciences, Universidad Santiago de Cali, Cali 760035, Colombia

**Keywords:** triazine, organic solar cells, dye-sensitized solar cells, perovskite materials, photovoltaics

## Abstract

Since the beginning of the 21st century, triazine-based molecules have been employed to construct different organic materials due to their unique optoelectronic properties. Among their applications, photovoltaics stands out because of the current need to develop efficient, economic, and green alternatives to energy generation based mainly on fossil fuels. Here, we review all the development of triazine-based organic materials for solar cell applications, including organic solar cells, dye-sensitized solar cells, and perovskite solar cells. Firstly, we attempt to illustrate the main synthetic routes to prepare triazine derivatives. Then, we introduce the main aspects associated with solar cells and their performance. Afterward, we discuss different works focused on the preparation, characterization, and evaluation of triazine derivatives in solar cells, distinguishing the type of photovoltaics and the role of the triazine-based material in their performance (e.g., as a donor, acceptor, hole-transporting material, electron-transporting material, among others). Throughout this review, the progress, drawbacks, and main issues of the performance of the mentioned solar cells are exposed and discussed. Finally, some conclusions and perspectives about this research topic are mentioned.

## 1. Introduction

One of the oldest known organic molecules, such that some of their derivative compounds have been known for at least 150 years, are 1,3,5-triazines, which are six-membered aromatic heterocyclic nitrogen derivatives. Initially, they were called symmetrical triazines, abbreviated as *s-* or *sym*-triazines, and nowadays, they are simply called triazines (so it will be used here until otherwise stated). Among the most important non-systematically named 1,3,5-triazines is cyanuric chloride, being used as a starting compound for synthesizing triazines, cyanuric acid, and melamine [1,2].

Chemist Ulric Nef first synthesized 1,3,5-triazine in 1895 through the reaction between hydrogen cyanide with ethanol in an ether solution saturated with hydrogen chloride. The resulting salt was treated with a base to produce an imidate intermediate, which produced 1,3,5-triazine. The product collection was carried out by distillation; however, they obtained a relatively low yield of 10%. In 1954, Grundmann and Kreutzberger modified Nef’s original method in which hydrogen chloride and hydrogen cyanide react to form 1,3,5-triazine through a 2,4,6-trichloro-1,3,5-triazine intermediate. They demonstrated by cryoscopic determination of the molecular weight that the compound was indeed the trimer, obtaining higher yield percentages [2,3,4]. Currently, triazines are synthesized in two ways: (i) by the nucleophilic displacement of the chlorine atom from cyanuric chloride in a staggered manner, or (ii) by the trimerization of nitriles, which is characterized by the direct formation of the triazine aromatic ring; resulting in a practical and straightforward method to prepare symmetric trisubstituted 1,3,5- or 2,4,6-triazines [5,6,7].

The molecular dimensions of triazine follow the general trend of other heterocyclic systems, having bonding angles in C-N-C less than 120°, which gives rise to a distorted hexagonal ring. The presence of substituents has a relatively small effect on bond lengths and ring angles [2]. Especially, 1,3,5-triazine is a π electron-deficient compound with a higher nucleophilic susceptibility compared to other six-membered heteroaromatic ring compounds with nitrogen atoms. These properties make triazine a useful moiety in organic synthesis and optical applications [5]. For example, triazine derivatives have been used as effective electron transporters in organic electronics due to their high electron affinity, optoelectronic properties, and thermal stability [8]. On the other hand, in the supramolecular chemistry field, they have been of great importance due to their *C_3_* symmetry and the related possibilities for the construction of several star-shaped molecules, porous materials, and molecular cages [9,10]. Furthermore, materials bound to a triazine core (an important π acceptor) have been evaluated as thermally activated delayed fluorescence dopants (TADF), showing good external quantum efficiencies using them, as well as suitable electron transport materials to obtain phosphorescent red and blue organic light-emitting diodes [11,12,13].

In this sense, highly conjugated molecules based on a triazine core have been widely synthesized and explored in the development of next-generation solar cells due to their ability to capture and transport electrons, as well as their low thermal conductivity [14]. Among these new-generation solar cells are organic solar cells (OSCs), dye-sensitized solar cells (DSSCs), and perovskite solar cells (PSCs), which have been of great interest and considered in the last decade for the development of energy alternatives to replace the current ones [15,16,17,18]. A great variety of hole transport materials (HTMs) based on a 1,3,5-triazine core have been designed for PSCs. For example, Triazine-Th-OMeTPA and Triazine-Ph-OMeTPA molecules were designed and presented the ability to stabilize the radical anion formed during light irradiation because of the presence of the triazine core. These molecules can replace the well-known spiro-OMeTAD, reaching energy conversion efficiencies (PCE) of 12.51% [19,20]. For DSSCs, a set of triazine-based dyes have been successfully synthesized containing bis(donor)-acceptor and donor-bis(acceptor) star-shaped molecular configurations. All of them exhibited absorption in the visible region with high molar extinction coefficients, compared to dyes without a triazine core and a maximum PCE of 4.29% [21]. In the case of OSCs, cathode interfacial materials (CIL) used in conventional and inverted OSCs (COSCs and IOSCs, respectively) have been prepared using triazine as a structural building block. Important examples of the above are 3-(4,6-bis(4-bromophenoxy)-1,3,5-triazin-2-yl)-2,6-difluorophenyl)diphenylphosphine oxide (Br-PO-TAZ) and 4,4’-((6-(3-(diphenylphosphoryl)-2,4-difluorophenyl)-1,3,5-triazine-2,4-diyl)bis(oxy)) dibenzonitrile (CN-PO-TAZ). The presence of a triazine core in those molecules has significantly improved the photovoltaic performance of the solar cells, obtaining PCEs of 8.19% and 8.33% for COSCs and IOSCs with CN-PO-TAZ, respectively. In contrast, Br-PO-TAZ produced a PCE of 8.15% in COSCs and 8.23% in IOSCs [22].

According to the above, this review aims to present and discuss the most relevant advances and applications of triazine-based organic materials for developing next-generation solar cells concisely, with a particular focus on PSCs, DSSCs, and OSCs. Essential aspects such as synthetic design, structural and physical properties, and performance of each discussed organic material are considered and approached in this work. In addition, some perspectives and potential points to be researched are included and discussed.

## 2. Common Synthetic Approaches to Obtain Substituted 1,3,5-Triazines

Different synthetic routes to afford substituted *s*-triazines have been proposed over time (see Figure 1). Some of them are based on obtaining or modifying a triazine core directly with identical or different substituents, producing symmetrical or asymmetrical substituted triazines, respectively [23,24]. Others modify substituted triazines to obtain larger and more sophisticated molecular structures [25]. The combination of both synthetic routes has led to the production of a large quantity of triazine with applications in different fields, including novel organic materials for applications in solar cells [22].

### 2.1. Direct Synthesis or Modification of a Triazine Core

Firstly, it is important to consider the electrophilic character of *s-*triazine; in comparison with benzene, triazine possesses lower delocalization energy and a smaller aromatic character, mainly due to the high polarity of the C=N bonds in the ring. One consequence of this is the high contribution of the polar canonical form (Figure 2) to the molecular resonance structure, where the carbon atoms have a deficient electron character. Thus, triazine results in being highly reactive against different types of nucleophiles (e.g., water and amines). However, these reactions do not produce nucleophilic substitution in the aromatic ring due to the pour leaving-group character of hydrogen atoms as a hydride. Instead, nucleophilic addition is followed by ring-opening processes leading to undesired products (see an example in Figure 1) [2,26].

The use of a highly electrophilic triazine derivative capable to suffer nucleophilic aromatic substitution in excellent yields is a better alternative to the above. Here, cyanuric chloride successfully fulfills this characteristic, and in conjunction with its low cost and high commercial availability, it is a widely used raw material to obtain substituted *s*-triazines [27]. Chemically, cyanuric chloride possesses chlorine atoms that serve as good leaving groups in nucleophilic aromatic substitution reactions, producing substituted triazines and hydrochloric acid in reaction with proton-donating nucleophiles. An advantage of this methodology is the control of nucleophilic substitution since the reactivity of each chlorine atom decreases as the substitution in the ring increases. In this way, the first substitution is usually carried out at 0 °C since it is exothermic. Subsequently, the second substitution can be performed at room temperature, and the third substitution is achieved at higher temperatures (>60 °C), as can be seen in Figure 2 [24,28]. Thus, it is possible to obtain trisubstituted *s*-triazines from cyanuric chloride simply just by carefully controlling the temperature in the reaction medium. In addition, different types of nucleophiles (e.g., amines, alcohols, thiols, among others) can be used as substituting agents along the process, resulting in many possibilities and triazine derivatives that can be evaluated and obtained through it [29].

Specifically, some important examples of the preparation of different substituted *s*-triazines from cyanuric chloride for applications in solar cells are reported in the literature [30,31,32,33,34,35]. In a synthetic sense, it has been highlighted that using cyanuric chloride as starting material can produce high yields. However, yield depends on different aspects, such as the type of nucleophile (i.e., its structure, basic strength, and steric factors), the nature of the solvent, and the presence of other substituents in the ring. As in another type of nucleophilic aromatic substitutions, it is expected that chlorine atoms in cyanuric chloride are easily substituted by reagents with a good basic character and low steric hindrance. For example, Liu and co-workers (2011) [34] employed a Grignard reagent and a small amine in nucleophilic aromatic substitutions on a triazine ring to obtain synthetic precursors (P1 and P2, in Figure 3) in the preparation of dyes with promising applications in dye-sensitized solar cells. The utilization of highly nucleophilic substrates allowed them to obtain substituted triazine in relatively high yields (>60%), under mild conditions, and in a simple way [33,34]. On the other hand, Mikroyannidis and colleagues (2010) [30] reported lower yields when they used a moderate nucleophile, such as 4-hydroxybenzaldehyde, to obtain a symmetrical substitute triazine (P3, in Figure 3) as a precursor of a low band gap organic molecule with application in organic solar cells. In this case, they employed the reflux condition to achieve the nucleophilic substitution of the third chlorine atom in the triazine ring.

In addition, bulky and electron-donating substituents tend to deactivate the ring against further nucleophilic substitutions, which results in lower yields. An example is the use of aminated bulky porphyrin fragments in nucleophilic substitution of cyanuric chloride, as reported by Sharma and co-workers (Figure 4) [35]. They proposed three consecutive steps of adding nucleophilic agents (two porphyrin derivatives and one amino acid methyl ester) in the same reaction medium with a gradual temperature and reaction time increase. Although this proposed strategy is coherent with previous observations and discussions on the reactivity of cyanuric chloride, yields not exceeding 40% were obtained in the preparation of new porphyrin-based photosensitizers for dye-sensitized solar cells [35].

On the other hand, the choice of solvent is more complex since it depends on solvent-reagent interactions (e.g., hydrogen bonds) and physical properties, such as boiling point and reagent solubility. For this, different types of organic solvents have been employed in these reactions, for example, dioxane, tetrahydrofuran, dimethylsulfoxide, and dichloromethane, among others [31,32,33,34,35,36]. In 2018, Sharma and co-workers [37] published work on studying cyanuric chloride’s reactivity to obtain substituted triazines. They employed a new concept called “orthogonal chemoselectivity” to describe the unique properties of cyanuric chloride as a raw material for obtaining substituted triazines, which is based on the discrimination among reactive sites in any order. In addition, they determined important structure-reactivity features in these reactions by comparing O^−^, N^−^, and S^−^nucleophiles, different solvents (tetrahydrofuran, dichloromethane, dimethylsulfoxide, and dioxane), and bases (K_2_CO_3_, triethylamine, and *N*,*N*-diisopropylethylamine). Generally, they observed a preferential order of incorporation of nucleophiles as alcohol > thiol > amine.

Another synthetic approach to obtaining substituted triazine cores is based on the cyclotrimerization of aliphatic and aromatic nitriles (Figure 5A). This reaction proceeds through a mechanism in which the carbon atom in each nitrile group is activated by a Lewis acid to suffer a further nucleophilic attack by the nitrogen atom of another nitrile group, resulting in a symmetrical or asymmetrical triazine ring after three steps of nucleophilic addition (Figure 5B) [23]. Different types of Lewis acids (e.g., ZnCl_2_, yttrium triflate, silica-supported Ti, silica-supported Zn, among others) [38,39] have been evaluated in the synthesis of substituted triazines by this route. However, harsh conditions (e.g., high temperatures) and long reaction times are necessary to obtain moderate to good yields (30–80%). At this point, microwave irradiation has been used to diminish reaction times, but higher temperatures are still required [40]. In addition, the relatively high cost of reagents and contaminant products are also drawbacks of this synthetic route. Thus, cyanuric chloride continues to be the most valuable starting material to obtain substituted triazines.

Alternatively, new substituted triazine cores have been obtained through other proposed routes, but some limitations in the protocols do not allow them to be widely used. For example, the reaction of cyanoguanidine with alkyl-, aryl-, and heteroarylnitriles has been used to obtain 6-substituted-2,4-diamino-1,3,5-triazines in high yields (52–91%) [41]. Alternatively, substituted triazines can be obtained from biguanides [42], imidic esters [43], aryl cyanates [44], and amidine salts [45], among others.

As mentioned previously, the leading synthetic route to obtain substituted *s-*triazines is the nucleophilic aromatic substitution of the chlorine atoms of cyanuric chloride. However, C-C coupling reactions between cyanuric chloride and alkyl or aryl compounds have also been used to obtain them. Among these, Suzuki and Stille’s reactions are two of the most representative cross-coupling reactions used for obtaining substituted triazines with a focus on solar cell applications [33,34,46,47]. In general, both types of reaction are classified as palladium-catalyzed cross-coupling reactions since they are based on the C-C coupling reaction of an organic halide with an organoboron (Suzuki cross-coupling) or organostannane (Stille cross-coupling) compound catalyzed by a palladium(0) complex (see Figure 6) [48,49]. Their particular mechanisms, employed reagents, and synthetical conditions make these reactions a powerful tool for efficient and versatile C-C bond formation, such that they have been applied in many research fields, including those with industrial purposes. Apart from Suzuki and Stille reactions, other types of cross-coupling reactions catalyzed by palladium(0) complexes have been developed through time (e.g., Heck coupling, Sonogashira coupling, Negishi coupling, among others), and they have their variants in terms of coupling agents, metallic complexes, and reaction conditions, but following the same catalytic principle (Figure 7) [49,50].

Several synthetic routes, including Suzuki and Stille’s couplings based on cyanuric chloride, have been reported in works focused on the preparation of substituted triazines for solar cell applications [30,32,33,34,47]. Among these, Liu et al. (2011) [34] employed in their synthetic route a Suzuki cross-coupling between 4-diphenylaminophenylboronic acid and a triazine derivative (Figure 8A) to obtain a precursor of triazine-based photosensitizers. In this reaction, they used a palladium(II) complex ([1,1′-bis(diphenylphosphino)ferrocene]dichloropalladium(II), abbreviated as PddppfCl_2_), which is reduced in situ to palladium(0) species to catalyze the reaction. Basic conditions, moderate temperatures, and short reaction times were sufficient to achieve a yield of 70%. In a similar work [33], the same authors used the cross-coupling between 4-formylphenylboronic acid and a disubstituted cyanuric chloride, using the same catalytic system, to obtain star-shaped triazine-based dyes in a 72% yield (Figure 8B). Likewise, Aryal and colleagues (2020) [22] coupled 2,4-difluorophenylboronic acid and cyanuric chloride through a Suzuki reaction, employing the well-known tetrakis (triphenylphosphine)palladium(0) (Pd(PPh_3_)_4_) as the catalyst (Figure 8C). The product was obtained with a yield of 45% and used to further functionalize by nucleophilic aromatic substitution of the remaining chlorine atoms in the ring. More recently, Duan and colleagues (2017) [47] used a Stille cross-coupling to obtain a trisubstituted triazine with perylene diimide groups for further application in organic solar cells. The reaction was performed between cyanuric chloride and a perylene diimide-based tributylstannane in the presence of Pd(PPh_3_)_4_ as the catalyst and CuI as the co-catalyst at 90 °C for 24 h (Figure 8D). A lower yield was obtained (36%) than the already-mentioned Suzuki couplings, but with a significant difference: three consecutive substitutions were carried out on the ring just by Stille cross-couplings.

### 2.2. Structural Modifications of Substituents of Triazine Cores

The second methodological approach is based on the modification of the substituents of the triazine core with the end of obtaining larger and more complex molecular structures with optimum physicochemical properties for the desired application. Different chemical reactions are possible and depend on the type of functional groups present in the substituents. Among these are condensation reactions, radical halogenations, palladium-catalyzed cross-couplings (including those where new C-N bonds are formed), and Horner–Wadsworth–Emmons (or Wittig–Horner) reactions that have been used in synthetic protocols focused on the preparation of triazine-based organic materials with promising application in solar cells. 

The first example of the above was given by Mikroyannidis and colleagues (2010) [30]. As was discussed previously, they obtained P3 by three consecutive nucleophilic substitutions of the chlorine atoms of cyanuric chloride (Figure 3C). After that, they employed the acidic character of the methylenic proton of 4-nitrobenzyl cyanide to attach it to the aldehyde group of P3 in anhydrous ethanol and basic conditions, obtaining the respective trisubstituted triazine with a more extensive conjugated π system (Figure 9). Importantly, this reaction gave high yields (85%) and represents a useful route to functionalize triazine substituents. A similar reaction was performed by Liu et al. to couple cyanoacetic acid with a triazine derivative, obtaining a yield of 69% in catalytic acid conditions [33].

On the other hand, the Horner–Wadsworth–Emmons reaction has also been used to extend the molecular structure of substituted triazines. This reaction is based on the nucleophilic addition of a phosphonate carbanion to a carbonyl compound (e.g., aldehyde) and the subsequent elimination of a dialkyl phosphate to produce an *E-*alkene predominantly. A particular example of this reaction is provided in Figure 10, where a triazine-based diphosphonate (produced by the reaction of a brominated agent with triethyl phosphite) is used to produce a substituted triazine with two donor fragments as arms for photovoltaic applications [34]. Strong basic conditions are employed in this reaction, and moderate-to-high yields are obtained [51,52].

Palladium-catalyzed cross-couplings are widely used reactions to modify the arms of substituted triazines. These reactions can be performed using halogenated triazine derivatives (e.g., brominated triazine derivatives) [21] or triazine-based organometallics (e.g., triazine-based boronate esters) [46]. For example, 2,4,6-*tris*(4-(4,4,5,5tetramethyl-1,3,2-dioxaborolan-2-yl)phenyl) [1,3,5] triazine is a joint starting compound to obtain large triazine-based structures by Suzuki couplings, as can be seen in Figure 11A [53]. In addition, the implementation of a tribrominated triazine compound serves as an appropriate synthetic route to produce star-shaped photoactive molecules by C-N cross-coupling (also called Buchwald–Hartwig amination) or Stille reactions. Thomas and co-workers combine these methods to obtain good yields (>80%) of different types of dyes for solar cell applications (Figure 11B) [21]. More recently, Zhang and colleagues (2021) [51] used 2,4-diphenyl-6-(4-(4,4,5,5-tetramethyl-1,3,2-dioxaborolan-2-yl)phenyl)-1,3,5-triazine to synthesize triazine-based photosensitizers by highly efficient Suzuki couplings. The products presented excellent optoelectronic properties for applications in dye-sensitized solar cells, as will be discussed later. 

## 3. Triazine and Its Derivatives in OSCs

### 3.1. Brief Introduction to OSCs

OSCs, also called “organic photovoltaics”, are one of the most studied and implemented alternatives to common inorganic solar cells (i.e., based on silicon and related). In general, OSCs use the electronic and optical properties of various organic materials, i.e., small organic molecules or organic polymers, for solar light harvest purposes. As these organic materials present high structural and chemical diversity, it is possible to modulate their electronic energy levels and, in this way, their optical band gaps, being one of the most important advantages of OSCs over inorganic solar cells. In addition, the organic materials implemented in OSCs can be easily obtained and have important properties such as large surface area and flexibility. This results in low-manufactured, versatile, and potentially efficient solar cells [54,55,56].

In contrast with silicon-based solar cells, OSCs are mainly constructed using two (or more) different organic materials with different optoelectronic properties, e.g., band gaps, in direct contact. One of them can promote electrons to a higher energy level when absorbing light (the donor), whereas the other can capture the electrons and move them through its matrix (the acceptor). This arrangement allows the generation of excitons, their transportation, and their separation into electrons and holes in an efficient way, such that it is possible to produce an electric current through the circuit with a relatively low energy input (0.5–1.0 eV) [57]. According to this, OSCs are fabricated in two architectures: bilayer heterojunction (BLHJ) and bulk heterojunction (BHJ) (see Figure 3). In this sense, triazine is an excellent building block for designing and constructing novel organic materials applied as promising acceptors in several architectures of OSCs. Some properties of the triazine molecule, such as high nucleophilic sensitivity, high electron affinity, and thermal stability, make it suitable for these applications, being possible to tune its optoelectronic properties by relatively simple structural modifications [22].

### 3.2. Triazine Derivatives in OSCs

As mentioned earlier, triazine has unique chemical, optical, and electronic properties, making it a good core for constructing different structural arrangements with excellent optoelectronic features. Among these arrangements, star-shaped conjugated molecules (SSCMs) are one of the most explored and applied triazine-based structures in solar cell materials and other kinds of optical devices (e.g., organic light-emitting diodes) [25,36,46]. The particular 3D structure of SSCMs can present a low susceptibility for the formation of ordered networks with high coplanar π-π stacking due to steric hindrance and dipole convergence phenomena, which results in improving notably charge transportation across the photoactive thin film. In addition, the optical and electronic properties of SSCMs are tunable simply by the modification of their core and arm units, as well as the π-bridge units and the conjugation system’s length, resulting in a diversity of possibilities to construct efficient photoactive organic materials [59]. Mainly, SSCMs are constructed with alternating donor and acceptor units in their structures, such that it is possible to tune their electronic properties, i.e., HOMO/LUMO relative energies or bandgap energy, depending on the type of employed unit. Thus, different combinations have been explored, for example, donor-acceptor-donor (D-A-D), acceptor-donor-acceptor (A-D-A), and their extended versions, where the central unit is the core of the star-shaped molecule [60]. The resulting small organic molecules have important advantages over the commonly employed photoactive materials, such as fullerenes and polymers, in terms of structural versatility, broad and isotropic absorption, as well as processability. For its part, fullerenes and their derivatives usually present weak light absorption, and it is difficult to modify their electronic properties, resulting in a limited performance. In contrast, polymers can be difficult to solubilize and process, making it complex to construct solar cells. This problem can be overcome by employing small organic molecules [61,62,63,64]. 

In this sense, various works focused on designing multifunctional triazine-based SSCMs, and their application in OSCs have been published through the years. Mainly, they have been used as efficient donors by the chemical modification of the electron-deficient triazine core with the donor’s arms. However, a few examples of triazine-based small molecules used as acceptors have also been reported. Figure 4 shows the chemical structures of some representative examples, while Table 1 summarizes their main optoelectronic properties and their performance when used in OSCs. Likewise, Figure 5 shows the structures of other materials mentioned in Table 1 and the discussion.

#### 3.2.1. Donor Materials

One of the first SSCMs based on a triazine core employed in an OSC was published in 2010 by Mikroyannidis and co-workers [30]. In this work, the authors proposed **A1** as a promising organic material for charge transport in a BHJ solar cell. For this, they obtained **A1** through a simple nucleophilic substitution of the three chloride atoms in cyanuric chloride with 4-hydroxybenzaldehyde in the presence of NaCO_3_. Subsequently, the intermediate product was subjected to a condensation reaction with 4-nitrobenzyl cyanide and NaOH in anhydrous ethanol to obtain **A1** in a high yield (85%). Structurally, the inclusion of electro-withdrawing terminal cyanovinylene 4-nitrophenyl groups aimed to extend the optical absorption of the resulting molecule. In addition, these groups, in conjunction with the ether linkages of **A1**, directly influenced its electronic levels. This way, **A1** presented an absorption spectrum from 300 to approximately 800 nm with a maximum of 648 nm, in both solution and thin film. The obtained E_g_^opt^ of 1.65 eV was smaller than the E_g_^opt^ of an alternating *p-*phenylenevinylene copolymer also synthesized by the authors, indicating that **A1** presented a stronger absorption ability. Likewise, the obtained HOMO/LUMO levels of −5.2 and −3.6 eV, respectively, were suitable for utilizing **A1** as a donor material in BHJ solar cells constructed employing PC_61_BM as an acceptor. Regarding efficiency, the BHJ solar cells based on **A1** presented a better performance than those based on the mentioned copolymer, obtaining a PCE close to 3.8 %. These results were associated with the relatively low bandgap, high hole mobility, and better quality of the film made from **A1**, highlighting the importance of employing a triazine core to design novel photoactive organic materials.

A few years later, Huang et al. [25] published work on designing and preparing new solution processable SSCMs based on a triazine core for BHJ solar cells, shown in Figure 4 and abbreviated as **A2** and **A3**. These were designed using triazine as the central acceptor unit, thienylenevinylene as π bridge units, and *tert*-butyl-substituted triphenylamine (tTPA) as the end of the arms and donor units. In terms of synthetic protocols, **A2** and **A3** were obtained through relatively simple Suzuki couplings employing a triazine-derivative boronate ester and the corresponding brominated donor fragments (or R groups of **A2** and **A3** shown in Figure 4), with acceptable yields (>30%). An essential characteristic of the obtained SSCMs was their thermal stability, which enables the processability of the molecules at high temperatures (even up to 400 °C without degradation) and makes them suitable for OSCs construction. In addition, UV-Vis characterization shows the strong absorption capability of the SSCMs due to an efficient intramolecular charge transfer (ICT) between the triazine core and tTPA branches, being greater for **A2** than **A3**. For the construction of BHJ solar cells, the authors employed two fullerene derivatives, PC_61_BM, and PC_71_BM, as the acceptor material and the synthesized SSCMs as the donor material. Both types of solar cells presented relatively low PCEs (0.20–2.48%), with a greater performance when using **A2**:PC_71_BM due to better light absorption in the visible region and charge transportation between donor and acceptor materials. The authors concluded that the evaluated triazine-based SSCMs represent a new, good chemical approach for designing new SSCMs for OSCs application.

Similarly, A-A-D structured SSCMs based on triazine as the central acceptor unit, 2,5-thienyl diketopyrrolopyrrole, and 1,4-phenylene diketopyrrolopyrrole as π bridge and acceptor units, tTPA and tert-butyl-substituted carbazole (tCz) as end groups and donor units (**A5**–**A7** in Figure 4) was reported by Shiau and co-workers in 2015 [46]. As in the previous example, the incorporation of donor units (tTPA and tCz) in the structure of triazine-based acceptor units aims the enhancement of ICT and light absorption by the molecules. Specifically, tTPA and tCz generate a redshift in the maximum absorption band of the SSCMs towards the visible region. For this, the authors employed two consecutive Suzuki couplings for obtaining **A5-A7** in relatively high yields (>60%). First, the coupling between the end groups and π bridges was done, followed by the coupling of the central triazine unit with the intermediate products. The photochemical properties of **A5**–**A7** are reported in Table 1. The λ_MAX_ at 623, 515, and 505 nm in solution for **A5**, **A6**, and **A7**, respectively, were associated with the ICT transitions between the donor units (tTPA or tCz) and π bridge units. In addition, strong ICT transitions associated with the triazine core of the molecules and the respective donor arms were also identified. This way, the prepared SSCMs were used as donor materials for OSCs fabrication with fullerene derivatives as acceptors due to their proper HOMO/LUMO levels (see Table 1). In bulk, both materials formed phase-separated interpenetrating networks with sizable domains, resulting in suitability for OSCs operation. In general, all SSCMs were successfully employed in BHJ solar cells; however, **A5** provided the highest photovoltaic performance (PCE > 1.5% with a photoactive layer of **A5**: PC_71_BM). This was associated with a lower energy band gap (1.58 eV) and a higher charge transfer capacity of **A5** compared to the others.

Other types of substituents have been used as the arms of the triazine core for preparing new donor materials. Interestingly, porphyrin is one of them, as seen in Figure 5. SSCMs based on triazine core and porphyrin fragments have been explored for BHJ solar cell construction by Sharma and co-workers since 2014 [31,36]. Porphyrin is a well-known macrocyclic structure capable of harvesting a high quantity of light, i.e., it has a broad and efficient absorption and carries out efficient electron transfer processes in biological systems [65]. Thus, it results in a suitable donor unit for the design of new SSCMs, with the possibility of changing the core unit and the metal cation present in the macrocyclic to tune their optoelectronic properties [66]. The first example of this approach is **A8**, obtained by Sharma et al. [36], through a nucleophilic substitution of the chloride atoms of cyanuric chloride by the amino groups of the zinc-metalated porphyrin derivatives and a simple secondary amine (piperidine). The resulting SSCM were characterized by UV-Vis spectroscopy and electrochemical measurements, obtaining satisfactory results in terms of strong and broad light absorption (λ_MAX_ at 425 nm, with a molar extinction coefficient of 6.66 × 10^5^ M^−1^ cm^−1^) and proper HOMO/LUMO energies (−5.69/–3.31 eV), which make them suitable for BHJ solar cell construction employing PC_71_BM as the electron acceptor material. Initially, the obtained PCE was 2.91% for the constructed solar cells, with a higher performance for **A8** than other discussed triazine-based SSCMs. However, the addition of pyridine (3% *v/v*) in the preparation of the photoactive layers led to the formation of a crystalline interpenetrating network with small domain sizes (10–20 nm), guaranteeing a more efficient exciton dissociation and more balance charge transport from **A8** to PC_71_BM. Through it, the PCE increased to 4.16%, with *V_OC_* of 0.94 V, *J_SC_* of 8.52 mA cm^−2^, and FF of 0.52. This approach indicated the importance of a good layer preparation step for obtaining a proper morphology for efficient free carrier generation and transportation.

Similarly, **A9** was prepared, characterized, and applied in a BHJ solar cell [31]. Mainly, **A9** contains three porphyrin fragments as donor units and an acceptor triazine core. The characterization methods showed similar absorption properties of **A9** in comparison with **A8**. However, although the photovoltaic performance of **A9**-based BHJ solar cells (using also PC_71_BM as an electron acceptor) was more significant (2.85–3.93%) than the photovoltaic performance of some already discussed SSCMs, it did not result to be greater than the performance of **A8**. Probably, this could be associated with a difference in the crystallinity degree and morphology of the photoactive blend materials.

#### 3.2.2. Acceptor Materials

Triazine-based SSCMs have also been used as non-fullerene acceptor materials in OSCs. Specifically, **A4** is a representative example of this. In 2017, Duan and colleagues [47] reported, in work titled “Pronounced Effects of a Triazine Core on Photovoltaic Performance–Efficient Organic Solar Cells Enabled by a PDI Trimer-Based Small Molecular Acceptor”, the preparation and application of **A4** in BHJ solar cells. For this, they synthesized the triazine-based SSCM in an acceptable yield (36%) using the so-called Stille reaction, which is based on the coupling of an organostannane (here, a perylene diimide tin derivative) with an organohalide (here, cyanuric chloride) in the presence of a palladium (0) catalyst and a copper(I) salt (e.g., CuI). Thermal, optical, and electrochemical properties of **A4** were evaluated in work, obtaining results consistent with a suitable acceptor material for OSCs fabrication: decomposition temperatures higher than 350 °C, strong and broad light absorption (390–620 nm, with molar extinction coefficients even up to 5.82 × 10^6^ M^−1^ cm^−1^), and HOMO/LUMO energies of –6.03/–3.81 eV. This way, **A4** was employed to fabricate a BHJ solar cell using PTB7-Th as electron donor material. When compared with a benzene-based analogous SSCM, **A4** presented a higher performance (a high PCE of 9.15% against 5.57% of the benzene derivative) due to the presence of a triazine core, which introduces beneficial effects for the molecular structure, e.g., broader and stronger absorption, better crystallinity, and higher free carrier transportation. The obtained PCE for **A4** is among the highest values reported for non-fullerene acceptor materials, which reflects the importance of a triazine core in SSCMs.

#### 3.2.3. Triazine Derivatives as Interfacial Layers for OSCs

Alternatively, triazine-based SSCMs have been applied as interfacial layers (ILs) for improving charge transportation and overall photovoltaic performance of OSCs, as well as for overcoming drawbacks associated with the air stability of some employed electrodes in those devices. More specifically, ILs can affect the energy level alignment, phase separation, and the local composition of the photoactive layer, which is in close contact with the IL [67,68]. In this sense, Chakravarthi and co-workers reported the preparation of a triazine-based small molecule with a phosphine oxide group (**A10**) and its application as an IL for improving the performance of OSCs [32]. In this work, the synthetic protocol was based on (i) a nucleophilic substitution reaction over the ring of cyanuric chloride, (ii) followed by a palladium-catalyzed cross-coupling, (iii) coupling of a lithiated intermediate product with chlorodiphenylphosphine, and (iv) an oxidation reaction of the final product to obtain the phosphine oxide group. In terms of operation, implementing a triazine core ensures good electron harvesting character, corroborated by ultraviolet photoelectron spectroscopy, obtaining deep HOMO/LUMO levels for **A10** (–6.96/–2.97 eV). In addition, the presence of a phosphine oxide group guarantees good solubility in environmentally friendly solvents, which makes the resulting molecules proper for large-area printing processes. This way, **A10** was employed as an IL in the construction of a BHJ solar cell with a photoactive layer of PBT7:PC_71_BM and two structures: one consisted of ITO/PEDOT:PSS/PBT7:PC_71_BM/**A10**/Al (also called conventional OSC), while the other was inverted and consisted in ITO/ZnO/**A10**/PBT7:PC_71_BM/PEDOT:PSS/Ag (called inverted OSC). The obtained performance was greater when using **PO-TAZ** as an IL for both types of OSCs, with an average PCE of 7.64 and 9.66% for conventional and inverted OSCs, respectively, in comparison with the performance of the solar cells without **A10**. This was associated with the good ability of this molecule for charge transporting between the electrode and photoactive material. In addition, it was noticed that the PCE was higher for inverted OSCs, which means that **A10** facilitates electron transfer with a better performance. Considering this, the authors evaluated another photoactive layer based on PBT7-Th/PC_71_BM with the implementation of **A10** as an IL, obtaining a high maximum PCE of 10.04% with values of *V_OC_* and *J_SC_* of 0.81 eV and 17.69 mA cm^−2^, respectively.

Recently, Aryal et al. [22]. Published a similar work focused on the preparation and application of two triazine-based small molecules (**A11** and **A12**) as an IL in BHJ solar cells. The molecules contained in their structures were a bromine atom and a cyano group as 4-substituents of the phenoxide rings, respectively, which were included for improving electron acceptor ability. As in the previous case, **A11** and **A12** exhibited an improvement in PCE between 1 to 2.5 percentage units for both conventional and inverted OSCs with a photoactive layer made from PBT7/PC_71_BM. Additionally, the fabricated ILs provided better air stability for the OSCs devices. The enhancement effects were attributed to essential properties of **A11** and **A12**, such as high charge transportation, reduced bulk resistance, reduced leakage current, and decreased series resistance.

## 4. Triazine and Its Derivatives in DSSCs

### 4.1. Brief Introduction to DSSCs

DSSCs are another type of new-generation solar cell with significant operational and economic properties. The first report on the design and implementation of DSSCs was published by O’Regan and Grätzel in 1991 [69]. In that first report, they constructed a solar cell using an electrode made from mesoporous TiO_2_ with a ruthenium complex (also called “dye sensitizer”) adsorbed on its surface, which could absorb light and inject electrons into the conduction band of the semiconductor. Likewise, they implemented a liquid electrolyte to restore the oxidized complex molecules through a redox reaction, completing a circuit with a proper electronic arrangement. This type of device-generated photocurrent with a PCE of 7–8%, being an important indication of the promising application of dye-sensitized electrodes in solar cells for green energy generation. Currently, DSSCs continue to be investigated due to their advantageous characteristics, such as good photovoltaic performances (even under low-light conditions), color and appearance versatility, simple fabrication, and low costs [70,71,72].

In general, the working principle and the main parts of a DSSC can be summarized and illustrated in Figure 6. Several key components in DSSCs determine their performance: the working electrode, the counter electrode, the redox mediator, and the type of photosensitizer. As with another type of solar cell, the aim is to optimize to a high degree the principal parameters of DSSCs to obtain the highest performance. Due to their particular operation principle, it is possible to optimize individual light absorption and charge transportation when dealing with DSSCs. According to this, the photosensitizer and redox mediator are the two main focuses for improving the performance of DSSCs [73].

Different types of redox mediators have been evaluated over time. The triiodide/iodide (I_3_^–^/I^–^) is the most used due to its efficiency. However, some limitations exist when using this redox mediator, such as volatilization, photodegradation, poor long-term stability, and electrode corrosion [74]. In this sense, other alternatives to the triiodide/iodide couple have been used. For example, ionic liquids, e.g., imidazolium-based ionic liquids, have emerged as eco-friendly redox mediators due to their low volatility and flammability, high chemical and thermal stability, moderate ionic conductivity, and proper redox behavior [75,76]. In addition, solid-state conducting materials such as inorganic and organic hole-transporting materials (HTMs) are promising alternatives to mediate charge transportation in DSSCs. These materials overcome the limitations associated with liquid redox couples and ionic liquids, such as complicated solar cell preparation and encapsulation, limited charge transportation, risk of leakage, and environmental contamination due to the usage of toxic organic solvents. Among different HTMs evaluated through time, those with an organic nature, e.g., spiro-OMeTAD, PEDOT, and triphenylamine-based HTMs, among others, have gained attention due to their easy preparation, tunable electrochemical properties, relatively low preparation costs, and effective deposition methods [77,78].

On the other hand, traditional dyes were based on ruthenium complexes with different kinds of organic and inorganic ligands, e.g., Black dye, N945, N3, and N719, among others. Important properties of these complexes make them useful in DSSCs, for example, long excited lifetimes, efficient metal-to-ligand charge transfer processes, and excellent redox behavior [79]. However, their costs, availability, and low extinction coefficients represent the most significant drawbacks of their use [70]. For this, organic metal-free dyes emerged as a promising alternative since they supply good advantages as photosensitizers, such as low preparation costs, high extinction coefficients, easy tunable electrochemical properties, structural diversity, and high stability. Generally, these dyes contain a high-conjugated π system with different donor and acceptor units by which it is possible to modulate their electrochemical properties [80,81]. One important example of organic dyes is those based on triazine, which use acceptor triazine units to construct structural arrangements with proper optoelectronic characteristics such that promising efficiencies have been obtained [33]. These will be discussed later.

### 4.2. Triazine-Based Photosensitizers in DSSCs

Different triazine-based photosensitizers or dyes with symmetrical and asymmetrical structures have been developed through time with a focus on their application in dye-sensitized solar cells. Figure 7 shows the chemical structures of the most representative examples of metal-free triazine-based dyes. Different types of substituents and tridimensional structures can be observed. Likewise, a particular class of dye based on Zn-metalated porphyrin units and triazine cores, which have gained attention in this research field, are also illustrated in Figure 8. These dyes will be discussed in a separate section. Table 2 and Table 3 summarize their main optoelectronic properties and performance characteristics in DSSCs.

#### 4.2.1. Metal-Free Triazine-Based Dyes

The first reported work focused on the preparation and application of triazine-based dyes for DSSCs was carried out by Liu and co-workers in 2011 [34]. In this research, they developed six dyes (**B1**–**B6**) based on a triazine as a central core linked to different units of donor and acceptor fragments. Suzuki cross-couplings and nucleophilic substitution directly on cyanuric chloride were the main reactions employed in their synthetic routes, obtaining relatively high yields (>65%). They determined interesting optical properties through absorption experiments, such as two absorption peaks at around 374–403 nm and 290–308 nm with molar absorptivity up to almost 78,900 M^−1^ cm^−1^. In those regions, it was observed that (i) the replacement of cyanoacetic acid (e.g., in **B1**) by rhodamine-3-acetic acid (e.g., in **B2**) led to an increase in absorption intensity and (i) the presence of triphenylamine fragments in the dyes gave; as a result, an increase in their molar absorptivities. The combination of these two effects allowed **B2** to exhibit excellent light absorption properties. Furthermore, all the dyes presented a higher LUMO energy than the energy level of TiO_2_, an indication of the ability to inject electrons into the conduction band of the semiconductor electrode. Finally, the performance of various DSSCs using the dyes was evaluated, obtaining acceptable results with *J_SC_* in the range from 0.96 to 3.33 mA·cm^−2^ and *V_OC_* from 0.510 to 0.757 V. The obtained values of PCE (0.35–1.81%) indicated that the dyes can be employed successfully in DSSCs.

Two years later (2013), the same authors [33] reported an extended work on the design of photosensitizers based on triazine as an acceptor core. In this case, they used larger conjugated fragments (including donor thiophene and furan groups in their structure) to obtain 13 new dyes with greater light absorption properties (**B7**–**B19**). The new dyes presented greater light absorption (ε up to 78,000 M^−1^ cm^−1^) and a red shift in their λ_MAX_ (up to 498 nm), resulting from a more conjugated structure and lower transition energies, possibly due to intramolecular charge transfers enhanced by thiophene and furan groups. For these dyes, the photovoltaic performance of the DSSCs was enhanced notably, reaching a *J_SC_* of 7.76 mA·cm^−2^ and a PCE of up to 3.69% for **B13**. This observation was explained considering the good properties of carboxyl groups to anchor to the semiconductor surface, which allows effective interaction between the dye and semiconductor, and the higher electron-injecting efficiency of **B13** into TiO_2_.

Thomas and co-workers [21] proposed the inclusion of fluorene and thiophene units in the structure of triazine-based dyes for improving their light-harvesting properties. To prove this, they synthesized three dyes (**B20**–**B22**) based on a triazine core with three fluorene substituents functionalized with carboxylic anchoring groups and donor diphenylamine and thiophene moieties. The dyes presented optimum HOMO and LUMO levels for appropriate electron injection and their regeneration after that process. Better light-harvesting properties were obtained for the mono-anchoring dyes (**B20** and **B21**) than for the bi-anchoring dye (**B22**). Especially, **B21** reached a PCE of 4.29%, attributed to a higher *J_SC_* (10.67 mA·cm^−2^), minimal charge recombination, and a high electron lifetime.

Other works have also studied the inclusion of thiophene fragments as donor arms in triazine-based dyes. For example, Zhang and colleagues [82] recently published work focused on preparing and evaluating three triazine dyes with conjugated thiophene units as their donor arms (**B23**−**B25**). In addition, they included alkyl groups in their arms to avoid dye aggregation onto the semiconductor surface and, in this sense, optimize the electron transport properties of the dyes. As a result, good optical properties were obtained for these dyes with λ_MAX_ between 373 and 478 nm and molar absorptivity up to 44,100 M^−1^ cm^−1^, which were also comparable to other triazine-based dyes mentioned previously [33,34]. In terms of electrochemical properties, the three dyes presented lower redox potentials in comparison with the commonly used redox couple I^−^/I_3_^−^ (0.4 V vs. NHE) and other reported photosensitizers, which indicates that the inclusion of a triazine unit can significantly reduce the HOMO energy level such that the regeneration of the dye can be carried out more easily. Next, a comparison was made between the performance of DSSCs constructed with each dye and in conjunction with a co-adsorbent (Chenodeoxycholic acid). The results suggested that the photosensitizers carry out an excellent performance in DSSCs, reaching a PCE higher than 7% (for **B23**) and electron mobility up to 14 mA·cm^−2^. Likewise, implementing a co-adsorbent can enhance the electronic behavior of the dyes in the solar cells, but a lower dye loading amount is obtained instead. It was noted that alkyl chains had a direct effect on the aggregation of the dye molecules so that the positionally dispersed alkyl groups (e.g., **B23**) suppress charge recombination and enhance the intra- and inter-molecular electron transport efficiency in a better way than the overly concentrated alkyl groups (e.g., **B24**).

#### 4.2.2. Zn-Metalated Porphyrin-Based Substituted Triazines

As discussed in a previous section, porphyrin-based structures can harvest a high quantity of light to present high absorption profiles and molar absorptivities. Likewise, it is possible to tune the optoelectronic properties of these structures by modification of the porphyrin structure with different types of substituents in meso or β positions [65,66]. Based on this approach, various works focused on the preparation of porphyrin-based dyes with triazine cores have been published [35,83,84,85], as can be seen in Figure 8 and Table 3. In general, all porphyrin-based dyes present a similar absorption profile, which is composed of a strong Soret band in 400–500 nm (hence, maximum absorption wavelengths are obtained at approx. 420 nm) and moderate Q bands in the 500–650 nm region [84]. This absorption is so high that molar absorptivities reach values higher than 600,000 M^−1^ cm^−1^, which are one order of magnitude greater than values for already discussed dyes. In addition, a triazine core is useful in these structures since it serves as a modulating agent for the energy of their frontier orbitals, resulting in dyes capable of injecting electrons into TiO_2_ and being regenerated by the electrolyte. Thus, DSSCs constructed with this type of dye can reach electron mobilities higher than 14 mA·cm^−2^ and PCE values higher than 5% and up to 7% (Table 3).

Several aspects affecting the performance of porphyrin/triazine-based dyes have been identified and highlighted: (i) aggregation in the semiconductor, which can be diminished by the modification of substituents in porphyrin and triazine rings, as well as by the utilization of a co-adsorbent in the solar cell preparation (e.g., CDCA) [84,85]; (ii) dye loading, which the inclusion of anchoring groups can optimize (e.g., carboxylic groups provided by an amino acid fragment) [83]; (iii) physical dimensions of mesoporous TiO_2_, which the preparation technique can modulate (e.g., paste coating or electrophoretic deposition) [85]; and (iv) charge recombination and electron lifetime, which can be suppressed by structure modification or employing co-sensitizers (e.g., D in Figure 9) [83].

**Table 3 molecules-28-00257-t003:** Optoelectronic properties and performance characteristics of Zn-metalated porphyrin-based triazines in DSSCs.

Triazine-Based Molecule	λ_MAX_ ^a^ (nm)	*ε*^b^ (×10^4^ M^−1^ cm^−1^)	E_ox_ ^c^ (V)	E_red_ ^d^ (V)	Redox Couple	*V_OC_* (V)	*J_SC_* (mA/cm^2^)	FF (%)	PCE (%)	Ref.
**C1**	422	-	1.16	−1.13	I^−^/I_3_^–^	0.63	3.33	65	3.61	[84]
**C2**	422	-	1.16	–0.89	I^–^/I_3_^–^	0.66	1.67	68	4.46	[84]
**C3**	422	81.7	0.92 ^g^	–1.16 ^g^	I^–^/I_3_^–^	0.63–0.66	9.43–10.94	64–68	3.80–4.91	[85]
**C3 (+CDCA)**	-	-	-	-	I^–^/I_3_^–^	0.64	12.42	70	5.56	[85]
**C4**	420	-	0.83	–1.33	I^–^/I_3_^–^	0.64	10.85	68	4.72	[83]
**C4 (+D)**	-	-	-	-	I^–^/I_3_^–^	0.70	14.78	71	7.34	[83]
**C5**	425	66.7	1.04	–1.08	I^–^/I_3_^–^	0.68	10.78	72	5.28	[35]
**C6**	425	66.6	1.29	–1.09	I^–^/I_3_^–^	0.62	8.56	63	3.50	[35]

^a^ In solution, ^b^ At λ_MAX_ in solution. ^c^ Versus normal hydrogen electrode (NHE). ^d^ E_red_ = E_ox_–E_0–0_, where E_0–0_ is the zeroth–zeroth transition energy. ^g^ Versus saturated calomel electrode (SCE). CDCA: Chenodeoxycholic acid. D: see its structure in Figure 9.

## 5. Triazine and Its Derivatives in PSCs

### 5.1. A Brief Introduction to PSCs

In 2009, Miyasaka and co-workers [86] published a work titled *Organometal Halide Perovskites as Visible-Light Sensitizers for Photovoltaic Cells* in the Journal of American Chemical Society. In this article, the authors reported the usage of organolead halide perovskite nanocrystals as efficient photosensitizers in DSSCs, since these materials presented unique and potentially useful optoelectronic properties which make them suitable light absorbers for photovoltaics, as determined in previous studies [87,88]. As in the case of any DSSC, the photosensitizer materials, i.e., organic-inorganic perovskites based on CH_3_NH_3_PbX_3_ (X=Br, I), were deposited on mesoporous TiO_2_ to serve as light absorbers, promoting electron transfer to the semiconductor. Using these perovskite materials, Miyasaka and colleagues could obtain PCEs of almost 4% in liquid DSSCs, a comparable or even higher value than some PCEs obtained in OSCs and other types of DSSCs at that time. This was the starting point of the development of PSCs.

Firstly, the research was focused on the effect of different parameters associated with the perovskite materials, e.g., the type of organic cation and particle size, and the construction of the solar cells, e.g., the electrolyte solvent and deposition method, reaching efficiencies between 2.4–6.5% until 2011 [89,90,91]. These limited efficiency values, the solubility of perovskite materials into the liquid electrolyte, and the instability of the resulting solar cells led to the need for designing more efficient and stable devices considering the inclusion of other materials and configurations. For instance, it was proposed to use an insulating layer made from Al_2_O_3_ between the perovskite-covered TiO_2_ and the electrolyte system, which allows the solar cell to be more stable against perovskite solubilization, recombination, and corrosion, as well as more efficient in comparison with liquid PSCs without Al_2_O_3_. A notable increase of PCE (from 3.56 to 6.00%) and a lower rate decay of *J_SC_* (more than 50% after 15 min) were observed [92]. However, the instability problem due to perovskite dissolution persisted, and the photocurrents generated by these devices were of short duration. To overcome this, the implementation of solid-state layers based on HTMs instead of liquid electrolyte systems was proposed [93]. The first HTM employed in PSCs based on CH_3_NH_3_PbI_3_ using mesoporous TiO_2_ was Spiro-MeOTAD. Surprisingly, the obtained PCE was 9.5% with stability of up to 500 h [94]. Undoubtedly, this confirmed the relevance of using solid-state HTMs in PSCs to enhance efficiency and stability. Further studies were focused on improving charge transport, mainly by employing dopant agents [95]. Thus, an increment of a few units in PCE values was achieved [96]. By 2012, Etgar and colleagues [97] reported the construction of an HTM-free PSC based on CH_3_NH_3_PbI_3_ and TiO_2_, obtaining a PCE of 5.5%, *J_SC_* of 16.1 mA/cm^2^, *V_OC_* of 0.631 V, and FF of 0.57, which showed a solar cell performance comparable to another type of photovoltaics. In principle, this revealed the dual behavior of perovskite materials: they can generate efficient charges in their bulk and, subsequently, transport them toward the respective terminal [98]. Likewise, this observation allowed us to modify and optimize the employed configurations of PSCs, which evolved into two main types: mesoporous and planar architectures (Figure 10). Thanks to this advance in solar cell configurations, higher efficiencies and stabilities were achieved, so currently, the maximum PCE obtained for PSCs is 25.7%, almost seven times greater than the first reported value [99].

Despite these advances in the efficiencies and stabilities of PSCs, researchers remain focused on improving various parameters of this type of solar cell, e.g., the structure of organic-inorganic perovskites and charge transporting materials, to reach their maximum performance. Among these parameters, charge-transporting layers are one of the main focuses due to the possibility of employing a multitude of organic materials, which can be structurally tunned to generate suitable properties for photovoltaic applications, such as easy processability, high stability, and proper electronic levels [91,92,93]. In this sense, many works focused on designing new organic materials, including small organic molecules and polymers, for charge transportation in PSCs have been published over time, which are approached in specialized reviews [100,101,102,103]. Here, we will primarily discuss those published works based on the design and synthesis of organic materials that possess triazine units and their use in PSCs as charge-transporting layers.

**Figure 10 molecules-28-00257-f010:**
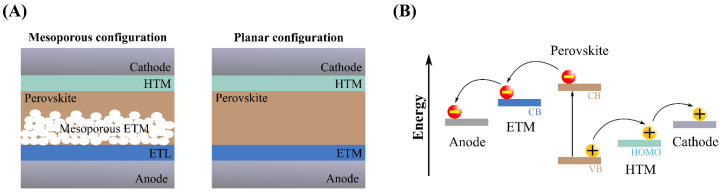
(**A**) Mesoporous and planar configurations of PSCs. (**B**) The energy diagram and charge transport in PSCs [102].

### 5.2. Triazine-Based Organic Materials in PSCs

Several triazine-based organic materials have been evaluated for different purposes in PSCs over time. These include star-shaped conjugated molecules for charge transportation (e.g., HTMs) [104], organic materials for surface passivation [105,106,107,108], and triazine derivatives for charge injection/extraction (e.g., as ILs) [109,110,111]. Among these, surface passivation and interfacial charge carriage are the main application of triazine-based organic materials in PSCs, as will be discussed next.

#### 5.2.1. Triazine Derivatives as HTMs

Two important examples of triazine derivative star-shaped molecules as HTMs in PCSs were reported by Ko and colleagues [104] in the middle of the last decade. They employed a triazine core as an acceptor unit coupled to different donor fragments, for example, triphenylamine derivatives. To efficiently couple and guarantee optimum charge transportation, the acceptor core and the donor fragments were coupled using various π-conjugated bridges, such as phenyl (**D1**), thiophene (**D2**), 9,9-dimethyl-9*H*-fluorene (**D3**), and 4,4-dimethyl-4*H*- indeno [1,2-*b*]thiophene (**D4**), as can be seen in Figure 11.

Initially, Ko et al. (2014) [19] reported the preparation and evaluation of **D1** and **D2** in PSCs as hole-transporting layers. They first synthesized the SSCMs from cyanuric chloride using Stille couplings with triphenylamine-based stannyl derivatives, obtaining yields of 50%. The resulting SSCMs presented HOMO energies that could match the HOMO level of the perovskite material, corroborating the potential capability of the triazine-based molecules to extract holes from the light-harvesting material. Considering the above, **D1** and **D2** were evaluated in mesoporous PSCs using TiO_2_ as the mesoporous ETL and MAPbI_3_ as the photoactive absorbed layer. The photovoltaic parameters of the PSCs are reported in Table 4. As observed, the solar cells based on **D1** and **D2** presented good photovoltaic performance, reaching PCEs higher than 10%. In particular, the SSCMs based on thiophene bridges (**D2**) presented better hole mobility (*μ*_h_) than those based on phenyl bridges (**D1**), which was associated with higher planarity in the molecule and, thus, a more effective π-conjugated system to transport charges. Likewise, the higher hole mobility of **D2** led to an improved fill factor, and PCE, which were comparable with those values obtained using Spiro-OMeTAD (PCE < 20%).

According to the above results, Ko and co-workers [104] were interested in studying and understanding the planarity-mobility relationships of triazine-based HTMs on the photovoltaic performances of PSCs. They evaluated the effect of other types of bridges in the structure of star-shaped triazine-based molecules, such as fluorene (**D3**) and indeno-thiophene derivatives (**D4**). Firstly, they employed a synthetic protocol based on Buchwald aminations to prepare triphenylamine-based donor fragments and consecutive Stille couplings to construct the star-shaped molecules with a triazine core, obtaining **D3** and **D4** with yields of 35–55%. The prepared molecules presented optimum optoelectronic properties to collect holes from the perovskite materials, i.e., MAPbI_3_. In this case, the PCE values were higher than 12% (see Table 4) and comparable with the performance obtained employing Spiro-OMeTAD as HTM. Important properties, such as thermal stability up to 400 °C and melting temperatures higher than 170 °C, were also reported. Specifically, the SSCMs based on the indeno-thiophene derivative bridges (**D4**) showed a higher conductivity (4.35 × 10^−4^ S cm^−1^) and a slower electron recombination rate, resulting in higher values of *V_OC_* and *J_SC_* in comparison with the values obtained for the SSCMs based on fluorene bridges (**D3**). Through this, the authors highlighted that (i) the conductivity of HTMs is one of the significant factors determining the performance of PSCs, (ii) it can be tunned just by a careful structural design, and (iii) triazine, as an acceptor core, represents an important starting point to generate organic materials with potential applications in PSCs.

#### 5.2.2. Triazine Derivatives as ILs for PSCs

Apart from the structural modification of perovskite materials and charge transporting materials, other approaches have been proposed to enhance the photovoltaic performance of PSCs. These approaches have been based on the improvement of the environmental stability of the solar cell components, such as the metal oxide ETM (TiO_2_ or ZnO), as well as the decrease in charge recombination at the surface of the charge conducting materials, which limits electron injection/extraction processes [109]. Thus, the use of ILs between the metal oxide and the photoactive perovskite layer has emerged as a solution to these limitations [55,67]. Different ILs have been proposed, including those based on organic materials made from triazine derivatives. Specifically, triazine-based ILs have been explored as efficient electron-transporting layers at the metal oxide’s surface and hydrophobic coatings to protect the metal oxide and perovskite material from moisture [112,113].

In 2016, Chakravarthi and co-workers [32] synthesized a triazine-based organic molecule to be used as ILs in PSCs. Specifically, they functionalized cyanuric chloride with two phenoxide fragments and one phosphine oxide fragment using nucleophilic substitution reactions and Suzuki cross-couplings, obtaining **E1** with an overall yield of 72%. The implementation of **E1** (see Figure 12) as ILs in planar PSCs based on MAPbI_3_ and ZnO as the photoactive and electron-transporting materials, respectively, led to an improvement of the photovoltaic performance (PCE of 16.2%) when compared with the solar cell without it (PCE of 13.6%), being a remarkable result at that time. This improvement was associated with the enhanced *J_SC_* and FF produced by a lower barrier for carrier extraction into the ETM. It has been highlighted that **E1** combines the polar effect of the phosphine oxide (P=O) group with the acceptor character of triazine, resulting in the formation of an interfacial dipole between the perovskite material and the ETM, which facilitates electron extraction/injection processes at the interface level. Likewise, a similar molecule (**E2**) was synthesized and evaluated two years later (2018) [111]. In this case, fluorine atoms were introduced at the *para* position of the phenoxide fragments to serve as ILs in constructing flexible planar PSCs based on ZnO as ETM. Besides causing an improvement in electron mobility (see Table 5) due to a lower work function of the cathode, the presence of fluorine atoms in the molecular structure of **E2** led to an increase in its glass transition temperature and thermal stability. This was traduced into ILs with easy processability and more robust morphology, diminishing any degradation or phase transition during operation. Furthermore, **E2** diminished the hydrophilic character of the ETM, as was confirmed by contact angle measurements, which ensures higher environmental stability.

Other triazine derivatives have been evaluated as ILs for improving PSCs, for example, **E3** and **E4** [109,110]. In these cases, one of the fragments corresponds to a Zn-metalated triphenylporphyrin (ZnTPP) derivative, which is introduced due to its notable optoelectronic properties, including its excellent electron transport. For instance, **E3** was used as ILs in planar PSCs based on TiO_2_ as ETM. To ensure a proper anchoring of **E3** onto the surface of the ETM, glycidyl fragments capable of coordinating Ti atoms were included in its structure. On the other hand, ZnTPP fragments, in conjunction with the triazine cores, were able to interact with the perovskite material (MAPbI_3_), extract electrons from it, and inject them into the ETM. The capability of **E3** to serve as ILs was observed in the improved values of *J_SC_* (23.8 mA cm^−2^) and PCE (16.9%) in comparison with those values of pure TiO_2_ (*J_SC_* and PCE of 21.3 mA·cm^−2^ and 15.0%, respectively). Another improvement was achieved in solar cell stability due to the hydrophobic character of the surface porphyrin layer over the metal oxide material. Similarly, **E4** was evaluated as an IL in planar PSCs. Particularly, this molecule possesses a boron-dipyrromethene (well-known as BODIPY) fragment that serves as a donor unit in conjunction with the ZnTPP fragment. When used as ILs, **E4** remarkably improved the photovoltaic performance of PSCs based on ZnO as ETM (PCE higher than 17%), which was an indication of its excellent capability to transfer electrons from the perovskite material to the metal oxide, possibly due to the “push-pull” character of the triazine-based porphyrin-bodipy derivative. Likewise, **E4** altered the nanomorphology of the metal oxide material in such a way that there was an enhancement of the photovoltaic performance and solar cell stability. These results confirm the effectivity of triazine derivatives to serve as ILs for photovoltaic purposes.

#### 5.2.3. Triazine Derivatives for Surface Passivation

Solution-processed perovskite materials are usually polycrystalline, which influences to a great extent, their optoelectronic properties and, in this sense, their performance when applied in photovoltaics. Specifically, perovskite materials generally have structural disorders at the surface level, such as grain boundary and crystallographic defects [114]. These surface defects can act as points of charge recombination and promote ion migration, resulting in a constant detriment of the PSCs and, consequently diminished performances [115]. In this sense, different additives have been employed for surface passivation of perovskite materials, including those based on triazine derivatives [105,106,107,108].

Relatively recent works focused on the use of triazine derivatives for surface passivation of PSCs have been published in the literature. Some examples are illustrated in Figure 13. For instance, Kim and co-workers [107] reported in 2018 the employment of melaminium iodide, the iodide salt of melamine (**F1**), to passivate the surface of the photoactive layer of mesoporous PSCs based on (FAPbI_3_)_0.875_(CsPbBr_3_)_0.125_, TiO_2_, and Spiro-OMeTAD. In particular, it was proposed that melaminium iodide can interact effectively with perovskite molecules through Lewis acid-based interactions (e.g., with the metal cation) and hydrogen bonds (e.g., with the organic cation), such that any detrimental effect of the photoactive layer due to ion migration or changes in morphology are diminished. This effect was corroborated by SLCL measurements, and the photovoltaic parameters obtained in PSCs using melaminium iodide (see Table 6). For example, a PCE value of up to 17.32% in conjunction with a diminished value of trap density (*n_t_*) was obtained, considering the control PSCs, i.e., without the iodide salt. Likewise, the presence of melaminium iodide generates a decrease in the hysteresis index (HI), being a positive indication of effective surface passivation by the compound. Furthermore, steady-state photoluminescence and time-resolved photoluminescence studies allowed the authors to determine that melaminium iodide produced faster charge separation at the perovskite/HTM interface. At the same time, no enhancement effect was appreciated at perovskite/ETM, possibly due to more favorable interactions with Spiro-OMeTAD than TiO_2_.

Further studies were carried out using organic molecules with more complex molecules, such as **F2**–**F6**. For example, Shi et al. [105] recently published a work focused on the surface passivation of PSCs by halogenated triphenyltriazines (**F2**–**F4**). As discussed in a previous section, these molecules were efficiently synthesized by cyclotrimerization of nitriles using trifluoromethanesulfonic acid as a reaction catalyst. When the molecules were used in PSCs based on MAPbI_3_ and mesoporous TiO_2_, remarkable results were obtained: (i) diminished trap density in comparison with the control, (ii) higher *J_SC_* and FF values, (iii) higher PCE values, and (iv) lower HI values (see Table 6). Specifically, fluorinated triphenyltriazine (**F2**) provided the best results, including the highest PCE value (19.81%) and the lowest HI (1.2%). According to its molecular structure, **F2** could stabilize the under-coordinate Pb^2+^ ions through π interactions with C=C and C=N bonds, resulting in efficient surface passivation. Also, positive results were obtained in solar cell stability, such that the constructed devices with **F2**–**F4** retained 73–81% of their PCE after 1200 h due to a lower moisture susceptibility.

Similarly, a perylene diimide-based triazine derivative (**F5**) was used for minimizing grain boundary defects and enhancing the coverage and crystal grain sizes of perovskite films in planar PSCs [108]. **F5** was added into the perovskite films of PSCs based on MAPbI_3_, a fullerene-based ETM, and PEDOT:PSS as HTM. The effect after **F5** addition in the PSC was the overall enhancement of its photovoltaic performance due to favorable interactions between the triazine derivative molecules and the perovskite material, which are traduced into the formation of larger grains, lower grain boundary defects, enhanced light absorption capacity of the perovskite film, and more hydrophobic character of the photoactive material. However, the PCE values obtained for this type of PSCs were limited compared to other exposed works.

A year later, Chen and colleagues (2020) [106] reported the incorporation of a nitrogen-rich two-dimensional material based on triazine-graphdiyne (**F6**) into planar PSCs to passivate surface defects of the photoactive material. Surprisingly, the results obtained were favorable in terms of performance enhancement and HI value reduction when **F6** was incorporated into MAPbI_3_- and FAPbI_3_-based PSCs, reaching PCE values of up to 21.2% and HI of 0.5%. Likewise, the authors highlighted the capability of **f6** to effectively passivate any grain boundary defects and reduce non-radiative recombination at the interface level without sacrificing conductivity due to favorable organic molecule-metal ion interactions and the electronic properties of the 2D material. This opened the possibility of considering other functional triazine-based materials for surface passivation in PSCs and ratified the importance of triazine derivatives for photovoltaic applications.

## 6. Conclusions and Perspectives

Many solar cells based on triazine derivatives have been explored and evaluated over time. It has been noted that triazine-based organic molecules possess important optoelectronic properties, such as high light-harvesting, fast charge transport, and good physicochemical stability, to accomplish several functions inside organic and hybrid solar cells, (e.g., donor or acceptor materials, photosensitizers, electron- or hole-transporting layers, interfacial materials, and surface passivation). Mainly, triazine moieties serve as acceptor units due to their electron-deficient nature, and through them, it is possible to modulate electronic energy levels in large molecular structures. To incorporate triazine cores in different molecules, e.g., SSCMs, two main approaches can be used: the modification of a triazine core or the modification of ring substituents. The former approach includes the use of reactions such as nucleophilic substitutions directly in cyanuric chloride, cyclotrimerization of nitriles, and palladium-catalyzed C-C cross-couplings. On the other hand, the second approach includes many reactions that, depending on the type of ring substituents and target structure, can be applied to obtain triazine-based molecules in high yields (>80%), e.g., carbanion-based condensations, Horner–Wadsworth–Emmons reaction, Suzuki cross-couplings, and Buchwald–Hartwig aminations. In this sense, it is valid to affirm that triazine represents a versatile building block to construct easily and efficiently functional organic materials.

After properly synthesizing the triazine-based organic material, it is used in OSCs and hybrid solar cells, i.e., DSSCs and PSCs. Triazine derivatives have been used as donor and acceptor materials in BHJ and BLHJ OSCs, photosensitizers in DSSCs, and HTMs in planar and mesoporous PSCs. For these types of photovoltaics, enhanced performances have been obtained when triazine-based organic materials have been included in their preparation. Specifically, PCE values between 7 and 10% have been reached in OSCs and DSSCs based on triazine derivatives, while PCE values even higher than 20% have been reported for PSCs with this type of organic material, which are competitive with those obtained using other, well-established organic materials. In general, it has been observed that triazine units enhance charge mobility in solar cells and, with it, their photovoltaic parameters. Apart from these applications, ILs and surface passivation have also been prepared from triazine cores, obtaining organic materials capable of participating in performance improvements, e.g., by facilitating interfacial charge transport, diminishing grain boundary defects in perovskite materials, and enhancing environmental stability.

Despite the significant progress in the application of triazine-based organic materials for photovoltaic purposes, some aspects need to be approached in future research, some of them the current drawbacks of this type of organic material to a proper application and compete with other alternatives: (i) exploring new molecular structures with tunned optoelectronic properties to serves efficiently as, for example, photosensitizers in DSSCs and charge-transporting materials in OSCs or PSCs; (ii) improving characteristics like electron injection/extraction, charge separation, redox behavior, and physicochemical stability through a careful design of the molecular structure; (iii) developing efficient and economical synthetic routes to prepare triazine-based organic materials; and (iv) evaluating the applicability of triazine derivatives in large-area devices. Considering the above, it is clear that triazine and its derivatives will continue to be part of a relevant research topic in organic synthesis and photovoltaics in the following years.

## Data Availability

Not applicable.

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
