# Peer review of "Triazine: An Important Building Block of Organic Materials for Solar Cell Application"

_molecules, 2022, doi:10.3390/molecules28010257_

Round 1

Reviewer 1 Report

Many thanks for submitting the review " Triazine: an important building block of organic materials for solar cell application" to Molecules. In this manuscript, the authors showed the main synthetic routes of triazine derivatives. Then, the manuscript introduced the main aspects associated with solar cells and their performance.  Subsequently, the paper discusses the different work in the preparation, characterization and evaluation of triazine derivatives in solar cells, the role of triazine materials as donor, acceptor, hole transport materials, electron transport materials and other properties.

 The application of triazine derivatives in solar cells as important skeleton structures of organic materials is discussed in this review. This review has collected a large amount of data, but it is only a simple description of the content reviewed, without comprehensive analysis, and too few of my own evaluation views. At the same time, the content of the review is too basic, and the introduction of organic solar cells, DSSC and PSC is too long. Since this article is a review of triazine derivatives, no space should be wasted on the basic knowledge of solar cells. In addition, too much is not well explained and the manuscript is far from ready for completion at this stage. I think a major revision should be made before publication in Molecules.

Also, there are still other problems to be addressed in this manuscript.

1. In part 3.2.2 of manuscript, compounds A10-A12 are used as the interface modification layer of organic solar cell devices, rather than as acceptor materials. The manuscript does not categorize the materials clearly.

2. In the manuscript, using the same shorthand for different phrases, such as interfacial layers (ILs) ionic liquids (ILs), will create ambiguity in the article.

Author Response

The authors thank and acknowledge their kind contribution to the review molecules-2099143, entitled: "Triazine: an important building block of organic materials for solar cell applications" in molecules.

We are convinced that all your contributions have help to improve the scientific quality of our review.

Reviewer 1

Many thanks for submitting the review " Triazine: an important building block of organic materials for solar cell application" to Molecules. In this manuscript, the authors showed the main synthetic routes of triazine derivatives. Then, the manuscript introduced the main aspects associated with solar cells and their performance.  Subsequently, the paper discusses the different work in the preparation, characterization and evaluation of triazine derivatives in solar cells, the role of triazine materials as donor, acceptor, hole transport materials, electron transport materials and other properties.

The application of triazine derivatives in solar cells as important skeleton structures of organic materials is discussed in this review. This review has collected a large amount of data, but it is only a simple description of the content reviewed, without comprehensive analysis, and too few of my own evaluation views. At the same time, the content of the review is too basic, and the introduction of organic solar cells, DSSC and PSC is too long. Since this article is a review of triazine derivatives, no space should be wasted on the basic knowledge of solar cells. In addition, too much is not well explained and the manuscript is far from ready for completion at this stage. I think a major revision should be made before publication in Molecules.

We included these suggestions in the text. The introduction of organic solar cells, DSSCs, and PSCs was modified, and the basic knowledge of solar cells was removed.

Also, there are still other problems to be addressed in this manuscript.

  1. In part 3.2.2 of manuscript, compounds A10-A12 are used as the interface modification layer of organic solar cell devices, rather than as acceptor materials. The manuscript does not categorize the materials clearly.

Response 1: it has been fixed. We added a new section titled “Triazine derivatives as interfacial layers for OSCs.”

  1. In the manuscript, using the same shorthand for different phrases, such as interfacial layers (ILs) ionic liquids (ILs), will create ambiguity in the article.

Point 2: This has been corrected. 

Reviewer 2 Report

Insuasty et al. focused on the development of triazine-based materials, especially including the application on organic solar cells, dye-sensitized solar cells and perovskite solar cells. The classification of the mentioned materials has been well addressed. However, the brief introduction of solar cells is not concise. For instance, the part 3.1.1 concerning the performance parameters is not necessary. Overall, I recommended this review for publication.

Author Response

The authors thank and acknowledge their kind contribution to the review molecules-2099143, entitled: "Triazine: an important building block of organic materials for solar cell applications" in molecules.

We are convinced that all your contributions have help to improve the scientific quality of our review.

Reviewer 2

Insuasty et al. focused on the development of triazine-based materials, especially including the application on organic solar cells, dye-sensitized solar cells and perovskite solar cells. The classification of the mentioned materials has been well addressed. However, the brief introduction of solar cells is not concise. For instance, the part 3.1.1 concerning the performance parameters is not necessary. Overall, I recommended this review for publication.

The introduction of solar cells has been modified. On the other hand, section 3.1.1 concerning basic knowledge of solar cells was removed as suggested.

Reviewer 3 Report

In this review, Braulio et al. describe the recent advances of triazine-based molecules for solar cell applications. The authors have discussed the different synthetic approaches to obtain substituted 1,3,5-triazines and their applications as electron acceptors in constructing various solar cells such as PSCs, DSSCs, and OSCs. In my view, the literature presented in this review is sufficient to warrant publication in Molecules. However, in view of this referee, the technical quality of the review has to be significantly improved prior to publication. In the following, I will mention just a few of the numerous flaws and inconsistencies I noted:

1)     All the solar cell parameters are well-defined in the literature. It does not improve the review quality by including these parameter definitions again. Please justify the inclusion of these parameter definitions.

2)     The schemes/chem draw figures are not uniform. Please provide the exact chem draw figures.

3)     Also, the quality of the figures is not good.

4)     The references are not uniform and adequately cited. 

Author Response

The authors thank and acknowledge their kind contribution to the review molecules-2099143, entitled: "Triazine: an important building block of organic materials for solar cell applications" in molecules.

We are convinced that all your contributions have help to improve the scientific quality of our review.

Reviewer 

In this review, Braulio et al. describe the recent advances of triazine-based molecules for solar cell applications. The authors have discussed the different synthetic approaches to obtain substituted 1,3,5-triazines and their applications as electron acceptors in constructing various solar cells such as PSCs, DSSCs, and OSCs. In my view, the literature presented in this review is sufficient to warrant publication in Molecules. However, in view of this referee, the technical quality of the review has to be significantly improved prior to publication. In the following, I will mention just a few of the numerous flaws and inconsistencies I noted:

1)     All the solar cell parameters are well-defined in the literature. It does not improve the review quality by including these parameter definitions again. Please justify the inclusion of these parameter definitions.

Response 1: Considering those parameters are well-defined in the literature, we decided to remove the basic knowledge of solar cells.

2)     The schemes/chem draw figures are not uniform. Please provide the exact chem draw figures.

Response 2: These have been corrected.

3)     Also, the quality of the figures is not good.

Response 3: These have been changed.

4)     The references are not uniform and adequately cited. 

Response 4: These have been corrected.

Round 2

Reviewer 1 Report

The manuscript has been sufficiently improved to warrant publication in Molecules.